# Learning long-range spatial dependencies with horizontal gated recurrent units

**Drew Linsley, Junkyung Kim, Vijay Veerabadran, Charlie Windolf, Thomas Serre**
Carney Institute for Brain Science
Department of Cognitive Linguistic & Psychological Sciences
Brown University
Providence, RI 02912
{drew_linsley,junkyung_kim,vijay_veerabadran,thomas_serre}@brown.edu

## Abstract

Progress in deep learning has spawned great successes in many engineering applications. As a prime example, convolutional neural networks, a type of feedforward neural networks, are now approaching – and sometimes even surpassing – human accuracy on a variety of visual recognition tasks. Here, however, we show that these neural networks and their recent extensions struggle in recognition tasks where co-dependent visual features must be detected over long spatial ranges. We introduce a visual challenge, Pathfinder, and describe a novel recurrent neural network architecture called the horizontal gated recurrent unit (hGRU) to learn intrinsic horizontal connections – both within and across feature columns. We demonstrate that a single hGRU layer matches or outperforms all tested feedforward hierarchical baselines including state-of-the-art architectures with orders of magnitude more parameters.

## 1 Introduction

Consider Fig. 1a which shows a sample image from a representative segmentation dataset [1] (left) and the corresponding contour map produced by a state-of-the-art deep convolutional neural network (CNN) [2] (right). Although this task has long been considered challenging because of the need to integrate global contextual information with inherently ambiguous local edge information, modern CNNs are capable to detect contours in natural scenes at a level that rivals that of human observers [2–6]. Now, consider Fig. 1b which depicts a variant of a visual psychology task referred to as "Pathfinder" [7]. Reminiscent of the everyday task of reading a subway map to plan a commute (Fig. 1c), the goal in Pathfinder is to determine if two white circles in an image are connected by a path. These images are visually simple compared to natural images like the one shown in Fig. 1a, and the task is indeed easy for human observers [7]. Nonetheless, we will demonstrate that modern CNNs struggle to solve this task.

Why is it that a CNN can accurately detect contours in a natural scene like Fig. 1a but also struggle to integrate paths in the stimuli shown in Fig. 1b? In principle, the ability of CNNs to learn such long-range spatial dependencies is limited by their localized receptive fields (RFs) – hence the need to consider deeper networks because they allow the buildup of larger and more complex RFs. Here, we use a large-scale analysis of CNN performance on the Pathfinder challenge to demonstrate that simply increasing depth in feedforward networks constitutes an inefficient solution to learning the long-range spatial dependencies needed to solve the Pathfinder challenge.

An alternative solution to problems that stress long-range spatial dependencies is provided by biology. The visual cortex contains abundant horizontal connections which mediate non-linear interactions between neurons across distal regions of the visual field [8, 9]. These intrinsic connections, popularly

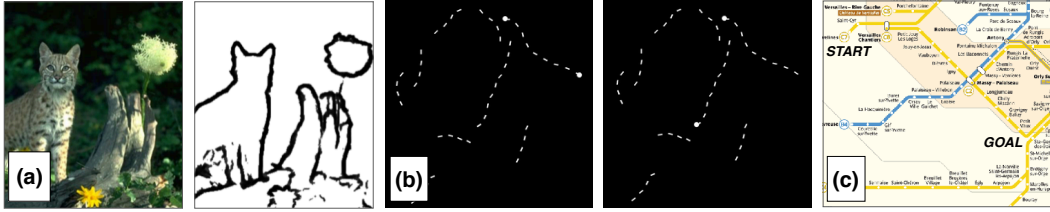

Figure 1: State-of-the-art CNNs excel at detecting contours in natural scenes, but they are strained by a task that requires the detection of long-range spatial dependencies. (a) Representative contour detection performance of a leading neural network architecture [23]. (b) Exemplars from the Pathfinder challenge: a task consisting of synthetic images which are parametrically controlled for long-range dependencies. (c) Long-range dependencies similar to those in the Pathfinder challenge are critical for everyday behaviors, such as reading a subway map to navigate a city.

called "association fields", are thought to form the main substrate for mechanisms of contour grouping according to Gestalt principles, by mutually exciting colinear elements while also suppressing clutter elements that do not form extended contours [10–15]. Such "extra-classical receptive field" mechanisms, mediated by horizontal connections, allow receptive fields to adaptively "grow" without additional processing depth. Building on previous computational neuroscience work [e.g., 10, 16–19], our group has recently developed a recurrent network model of classical and extra-classical receptive fields that is constrained by the anatomy and physiology of the visual cortex [20]. The model was shown to account for diverse visual illusions providing computational evidence for a novel canonical circuit that is shared across visual modalities.

Here, we show how this computational neuroscience model can be turned into a modern end-to-end trainable neural network module. We describe an extension of the popular gated recurrent unit (GRU) [21], which we call the horizontal GRU (hGRU). Unlike CNNs, which exhibit a sharp decrease in accuracy for increasingly long paths, we show that the hGRU is highly effective at solving the Pathfinder challenge with just *one layer* and a fraction of the number of parameters and training samples needed by CNNs. We further find that, when trained on natural scenes, the hGRU learns connection patterns that coarsely resemble anatomical patterns of horizontal connectivity found in the visual cortex, and exhibits a detection profile that strongly correlates with human behavior on a classic contour detection task [22].

**Related work**    Much previous work on recurrent neural networks (RNNs) has focused on modeling sequences with learnable gates in the form of long-short term memory (LSTM) units [24] or gated recurrent units (GRUs) [21]. RNNs have also been extended to learning spatial dependencies in static images with broad applications [25–29]. In this approach, images are transformed into one-dimensional sequences that are used to train an RNN. In recent years, several approaches have introduced convolutions into RNNs, using the recursive application of convolutional filters as a method for increasing the depth of processing through time on tasks like object recognition and super-resolution without additional parameters [30–32]. Other groups have constrained these convolutional-RNNs with insights from neuroscience and cognitive science, engineering specific patterns of connectivity between processing layers [33–36]. The proposed hGRU builds on this line of biologically-inspired implementations of RNNs, adding connectivity patterns and circuit mechanisms that are typically found in computational neuroscience models of neural circuits [e.g., 10, 16–20].

Another class of models related to our proposed approach is Conditional Random Fields (CRFs), probabilistic models aimed at explicitly capturing associations between nearby features. The connectivity implemented in CRFs is similar to the horizontal connections used in the hGRU, and has been successfully applied as a post-processing stage in visual tasks such as segmentation [37, 38] to smooth out and increase the spatial resolution of prediction maps. Recently, such probabilistic methods have been successfully incorporated in a generative vision model shown to break text-based CAPTCHAs [39]. Originally formulated as probabilistic models, CRFs can also be cast as RNNs [40].

## 2 Horizontal gated recurrent units (hGRUs)

**Original contextual neural circuit model**    We begin by referencing the recurrent neural model of contextual interactions developed by Mély et al. [20]. Below we adapted the model notations to a computer vision audience. Model units are indexed by their 2D positions $(x, y)$ and feature channel $k$. Neural activity is governed by the following differential equations (see Supp. Material for the full treatment):

$$\eta \dot{H}^{(1)}_{xyk} + \epsilon^2 H^{(1)}_{xyk} = \left[ \xi X_{xyk} - (\alpha H^{(1)}_{xyk} + \mu) C^{(1)}_{xyk} \right]_+$$
$$\tau \dot{H}^{(2)}_{xyk} + \sigma^2 H^{(2)}_{xyk} = \left[ \gamma C^{(2)}_{xyk} \right]_+ . \tag{1}$$

where

$$C^{(1)}_{xyk} = (\mathbf{W}^I * \mathbf{H}^{(2)})_{xyk}$$
$$C^{(2)}_{xyk} = (\mathbf{W}^E * \mathbf{H}^{(1)})_{xyk},$$

Here, $\mathbf{X} \in \mathbb{R}^{W \times H \times K}$ is the feedforward drive (i.e., neural responses to a stimulus), $\mathbf{H}^{(1)} \in \mathbb{R}^{W \times H \times K}$ is the recurrent circuit input, and $\mathbf{H}^{(2)} \in \mathbb{R}^{W \times H \times K}$ the recurrent circuit output. Modeling input and output states separately allows for the implementation of a particular form of inhibition known as "shunting" (or divisive) inhibition. Unlike the excitation in the model which acts linearly on a unit's input, inhibition acts on a unit's output and hence, regulates the unit response non-linearly (i.e., given a fixed amount of inhibition and excitation, inhibition will increase with the unit's activity unlike excitation which will remained constant).

The convolutional kernels $\mathbf{W}^I, \mathbf{W}^E \in \mathbb{R}^{S \times S \times K \times K}$ describe inhibitory vs. excitatory hypercolumn connectivity (constrained by anatomical data[1]). The scalar parameters $\mu$ and $\alpha$ control linear and quadratic (i.e., shunting) inhibition by $\mathbf{C}^{(1)} \in \mathbb{R}^{W \times H \times K}$, $\gamma$ scales excitation by $\mathbf{C}^{(2)} \in \mathbb{R}^{W \times H \times K}$, and $\xi$ scales the feedforward drive. Activity at each stage is linearly rectified (ReLU) $[\cdot]_+ = \max(\cdot, 0)$. Finally, $\eta, \epsilon, \tau$ and $\sigma$ are time constants. To make this model amenable to modern computer vision applications, we set out to develop a version where all parameters could be trained from data. If we let $\eta = \tau$ and $\sigma = \epsilon$ for symmetry and apply Euler's method to Eq. 1 with a time step of $\Delta t = \eta/\epsilon^2$, then we obtain the discrete-time equations:

$$H^{(1)}_{xyk}[t] = \epsilon^{-2} \left[ \xi X_{xyk} - (\alpha H^{(1)}_{xyk}[t-1] + \mu) C^{(1)}_{xyk}[t] \right]_+$$
$$H^{(2)}_{xyk}[t] = \epsilon^{-2} \left[ \gamma C^{(2)}_{xyk}[t] \right]_+ . \tag{2}$$

Here, $\cdot[t]$ denotes the approximation at the $t$-th discrete timestep. This results in a trainable convolutional recurrent neural network (RNN) which performs Euler integration of a dynamical system similar to the neural model of [20].

**hGRU formulation**    We build on Eq. 2 to introduce the hGRU – a model with the ability to learn complex interactions between units via horizontal connections within a single processing layer (Fig. 2). The hGRU extends the derivation from Eq. 2 with three modifications that improve the training of the model with gradient descent and its expressiveness[2]. (i) We introduce learnable gates, borrowed from the gated recurrent unit (GRU) framework (see Supp. Material for the full derivation from Eq. 2). (ii) The hGRU makes the operations for computing $\mathbf{H}^{(2)}$ (excitation) symmetric with those of $\mathbf{H}^{(1)}$ (inhibition), providing the circuit the ability to learn how to implement linear and quadratic interactions at each of these processing stages. (iii) To control unstable gradients, the hGRU uses a squashing pointwise non-linearity and a learned parameter to globally scale activity at every processing timestep (akin to a constrained version of the recurrent batchnorm [41]).

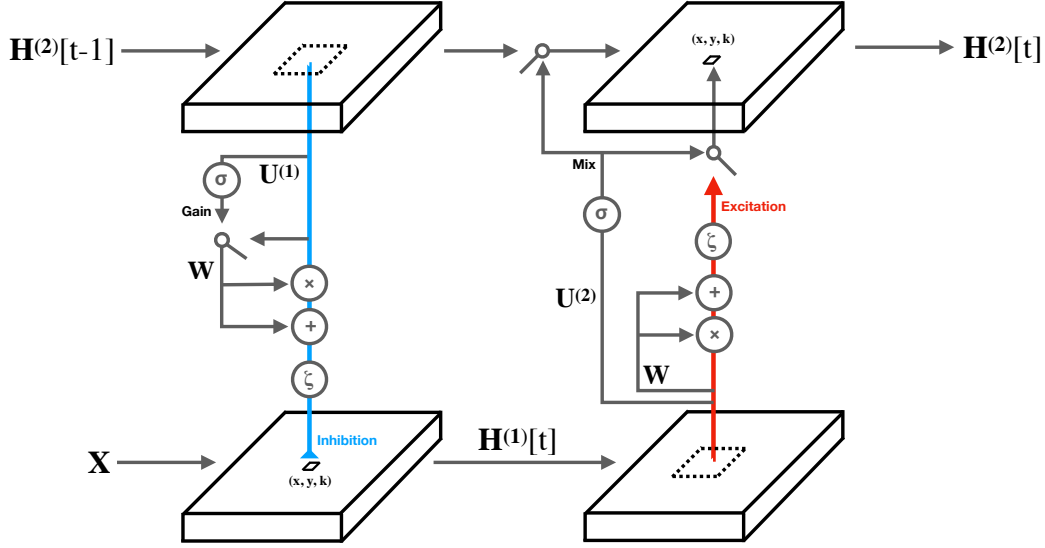

Figure 2: The hGRU circuit. The hGRU can learn highly non-linear interactions between spatially neighboring units in the feedforward drive $\mathbf{X}$, which are encoded in its hidden state $\mathbf{H}^{(2)}$. This computation involves two stages, which are inspired by a recurrent neural circuit of horizontal connections [20]. First, the horizontal inhibition (blue) is calculated by applying a gain to $\mathbf{H}^{(2)}[t-1]$, and convolving the resulting activity with the kernel $W$ which characterizes these interactions. Linear ($+$ symbol) and quadratic ($\times$ symbol) operations control the convergence of this inhibition onto $\mathbf{X}$. Second, the horizontal excitation (red) is computed by convolving $\mathbf{H}^{(1)}[t]$ with $W$. Another set of linear and quadratic operations modulate this activity, before it is mixed with the persistent hidden state $\mathbf{H}^{(2)}[t-1]$. Note that the excitation computation involves an additional "peephole" connection, not depicted here. Small solid-line squares within the hypothetical activities that the circuit operates on denote the unit indexed by 2D position $(x, y)$ and feature channel $k$, whereas dotted-line squares depict the unit's receptive field (a union of both classical and extra-classical definitions) in the previous activity.

In our hGRU implementation, the feedforward drive $\mathbf{X}$ corresponds to activity from a preceding convolutional layer. The hGRU encodes spatial dependencies between feedforward units via its (time-varying) hidden states $\mathbf{H}^{(1)}$ and $\mathbf{H}^{(2)}$. Updates to the hidden states are managed using two activities, referred to as the reset and update "gates": $\mathbf{G}^{(1)}$ and $\mathbf{G}^{(2)}$. These activities are derived from convolutions, denoted by $*$, between the kernels $\mathbf{U}^{(1)}, \mathbf{U}^{(2)} \in \mathbb{R}^{1 \times 1 \times K \times K}$ and hidden states $\mathbf{H}^{(1)}$ and $\mathbf{H}^{(2)}$, shifted by biases $\mathbf{b}^{(1)}, \mathbf{b}^{(2)} \in \mathbb{R}^{1 \times 1 \times K}$, respectively. The pointwise non-linearity $\sigma$ is applied to each activity, normalizing them in the range $[0, 1]$. Because these activities are real-valued, we hereafter refer to the reset gate as the "gain", and the update gate as the "mix".

Horizontal interactions between units are calculated by the kernel $\mathbf{W} \in \mathbb{R}^{S \times S \times K \times K}$, where $S$ describes the spatial extent of these connections in a single timestep (Fig. 2; but see Supp. Material for a version with separate kernels for excitation vs. inhibition, as in Eq. 2). Consistent with computational models of neural circuits (e.g., [10, 16–20]), $\mathbf{W}$ is constrained to have symmetric weights between channels, such that the weight $W_{x_0+\Delta x, y_0+\Delta y, k_1, k_2}$ is equal to the weight $W_{x_0+\Delta x, y_0+\Delta y, k_2, k_1}$ where $x_0$ and $y_0$ denote the center of the kernel. This constraint reduces the number of learnable parameters by nearly half vs. a normal convolutional kernel. Hidden states $\mathbf{H}^{(1)}$ and $\mathbf{H}^{(2)}$ are recomputed via horizontal interactions at every timestep $t \in [0, T]$. We begin by describing computation of $\mathbf{H}^{(1)}[t]$:

$$\mathbf{G}^{(1)}[t] = \sigma(\mathbf{U}^{(1)} * \mathbf{H}^{(2)}[t-1] + \mathbf{b}^{(1)}) \tag{3}$$

$$C^{(1)}_{xyk}[t] = (\mathbf{W} * (\mathbf{G}^{(1)}[t] \odot \mathbf{H}^{(2)}[t-1]))_{xyk} \tag{4}$$

$$H^{(1)}_{xyk}[t] = \zeta(X_{xyk} - C^{(1)}_{xyk}[t](\alpha_k H^{(2)}_{xyk}[t-1] + \mu_k)) \tag{5}$$

Channels in $\mathbf{H}^{(2)}[t-1]$ are first modulated by the gain[3] $\mathbf{G}^{(1)}[t]$. The resulting activity is convolved with $\mathbf{W}$ to compute $\mathbf{C}^{(1)}[t]$, which is the horizontal inhibition of the hGRU at this timestep. This inhibition is applied to $\mathbf{X}$ via the parameters $\boldsymbol{\mu}$ and $\boldsymbol{\alpha}$, which are $k$-dimensional vectors that respectively scale linear and quadratic (akin to shunting inhibition described in Eq. 1) terms of the horizontal interaction with $\mathbf{X}$. The pointwise $\zeta$ is a hyperbolic tangent that squashes activity into the range $[-1, 1]$ (but see Supp. Material for a hGRU with a rectified linearity). Importantly, in contrast to the original circuit, in this formulation the update to $\mathbf{H}^{(1)}[t]$ (Eq. 5) is calculated by combining horizontal connection contributions of $\mathbf{C}^{(1)}[t]$ with $\mathbf{H}^{(2)}[t-1]$ rather than $\mathbf{H}^{(1)}[t-1]$, which we found improved learning on the visual tasks explored here.

The updated $\mathbf{H}^{(1)}[t]$ is next used to calculate $\mathbf{H}^{(2)}[t]$.

$$G_{xyk}^{(2)}[t] = \sigma((\mathbf{U}^{(2)} * \mathbf{H}^{(1)}[t])_{xyk} + b_k^{(2)}) \tag{6}$$

$$C_{xyk}^{(2)}[t] = (\mathbf{W} * \mathbf{H}^{(1)}[t])_{xyk} \tag{7}$$

$$\tilde{H}_{xyk}^{(2)}[t] = \zeta(\kappa_k H_{xyk}^{(1)}[t] + \beta_k C_{xyk}^{(2)}[t] + \omega_k H_{xyk}^{(1)}[t] C_{xyk}^{(2)}[t]) \tag{8}$$

$$H_{xyk}^{(2)}[t] = \eta_t(H_{xyk}^{(2)}[t-1](1 - G_{xyk}^{(2)}[t]) + \tilde{H}_{xyk}^{(2)}[t] G_{xyk}^{(2)}[t]) \tag{9}$$

The mix $\mathbf{G}^{(2)}[t]$ is calculated by convolving $\mathbf{U}^{(2)}[t]$ with $\mathbf{H}^{(1)}[t]$, followed by the addition of $\mathbf{b}^{(2)}$. The activity $\mathbf{C}^{(2)}[t]$ represents the excitation of horizontal connections onto the newly-computed $\mathbf{H}^{(1)}[t]$. Linear and quadratic contributions of horizontal interactions at this stage are controlled by the $k$-dimensional parameters $\boldsymbol{\kappa}$, $\boldsymbol{\omega}$, and $\boldsymbol{\beta}$. The parameters $\boldsymbol{\kappa}$ and $\boldsymbol{\omega}$ control the linear and quadratic contributions of horizontal connections to $\tilde{\mathbf{H}}^{(2)}[t]$. The parameter $\boldsymbol{\beta}$ is a gain applied to $\mathbf{C}^{(2)}[t]$, giving $\mathbf{W}$ an additional degree of freedom in expressing this excitation. With this full suite of interactions, the hGRU can in principle implement both a linear and a quadratic form of excitation (i.e., to assess self-similarity), each of which play specific computational roles in perception [42]. Note that the inclusion of $\mathbf{H}^{(1)}[t]$ in Eq. 8 functions as a "peephole" connection between it and $\tilde{\mathbf{H}}^{(2)}[t]$. Finally, the mix $\mathbf{G}^{(2)}$ integrates the candidate $\tilde{\mathbf{H}}_t^{(2)}$ with $\mathbf{H}_t^{(2)}$. The learnable $T$-dimensional parameter $\boldsymbol{\eta}$, which we refer to as a time-gain, helps control unstable gradients during training. This time-gain modulates $\mathbf{H}_t^{(2)}$ with the scalar, $\eta_t$, which as we show in our experiments below improves model performance.

## 3   The Pathfinder challenge

We evaluated the limits of feedforward and recurrent architectures on the "Pathfinder challenge", a synthetic visual task inspired by cognitive psychology [7]. The task, depicted in Fig. 1b, involves detecting whether two circles are connected by a path. This is made more difficult by allowing target paths to curve and introducing multiple shorter unconnected "distractor" paths. The Pathfinder challenge involves three separate datasets, for which the length of paths and distractors are parametrically increased. This challenge therefore screens models for their effectiveness in detecting complex long-range spatial relationships in cluttered scenes.

**Stimulus design**   Pathfinder images were generated by placing oriented "paddles" on a canvas to form dashed paths. Each image contained two paths made of a fixed number of paddles and multiple distractors made of one third as many paddles. Positive examples were generated by placing two circles at the ends of a single path (Fig. 1b, left) and negative examples by placing one circle at the end of each of the paths (Fig. 1b, right). The paths were curved and variably shaped, with the possible number of shapes exponential to the path length. The Pathfinder challenge consisted of three datasets, in which path and distractor length was successively increased, and with them, the overall task difficulty. These datasets had path lengths of 6, 9 and 14 paddles, and each contained 1,000,000 unique images of 150×150 pixels. See Supp. Material for a detailed description of the stimulus generation procedure.

**Model implementation**    We performed a large-scale analysis of the effectiveness of feedforward and recurrent computations on the Pathfinder challenge. We controlled for the effects of idiosyncratic model specifications by using a standard architecture, consisting of "input", "feature extraction", and "readout" processing stages. Swapping different feedforward or recurrent layers into the feature extraction stage let us measure the relative effectiveness of each on the challenge. All models except for state-of-the-art "residual networks" (ResNets) [43] and per-pixel prediction architectures were embedded in this architecture, and these exceptions are detailed below. See Supp. Material for a detailed description of the input and readout stage. Models were trained on each Pathfinder challenge dataset (Fig. 3d), with 90% of the images used for training (900,000) and the remainder for testing (100,000). We measured model performance in two ways. First, as the accuracy on test images. Second, as the "area under the learning curve" (ALC), or mean accuracy on the test set evaluated after every 1000 batches of training, which summarized the rate at which a model learned the task. Accuracy and ALC were taken from the model that achieved the highest accuracy across 5 separate runs of model training. All models were trained for two epochs except for the ResNets, which were trained for four. Model training procedures are detailed in Supp. Material.

**Recurrent models**    We tested 6 different recurrent layers in the feature extraction stage of the standard architecture: hGRUs with 8, 6, and 4-timesteps of processing; a GRU; and hGRUs with lesions applied to parameters controlling linear or quadratic horizontal interactions. Both the GRU and lesioned versions of the hGRU ran for 8 timesteps. These layers had $15 \times 15$ horizontal connection kernels ($W$) with an equal number of channels as their input layer (25 channels).

We observed 3 overarching trends: First, each model's performance monotonically decreased, or "strained", as path length increased. Increasing path length reduced model accuracy (Fig. 3a), and increased the number of batches it took to learn a task (Fig. 3b). Second, the 8-timestep hGRU was more effective than any other recurrent model, and it outperformed each of its lesioned variants as well as a standard GRU. Notably, this hGRU was strained the *least* by the Pathfinder challenge out of all tested models, with a negligible drop in accuracy as path length increased. This finding highlights the effectiveness of the hGRU for processing long-range spatial dependencies, and how the dynamics implemented by its linear and quadratic horizontal interactions are important. Third, hGRU performance monotonically decreased with processing time. This revealed a minimum number of timesteps that the hGRU needed to solve each Pathfinder dataset: 4 for the length-6 condition, 6 for the length-9 condition, and 8 for the length-14 condition (first vs. second columns in Fig. 3a). Such time-dependency in the Pathfinder task is consistent with the accuracy-reaction-time tradeoff found in humans as the distance between endpoints of a curve increases [7].

**Feedforward models**    We screened an array of feedforward models on the Pathfinder challenge. Model performance revealed the importance of kernel size vs. kernel width, model depth, and feedforward operations for incorporating additional scene context for solving Pathfinder. Model construction began by embedding the feature extraction stage of the standard model with kernels of one of three different sizes: $10 \times 10$, $15 \times 15$, or $20 \times 20$. These are referred to as small, medium, and large kernel models (Fig. 3). To control for the effect of network capacity on performance, the number of kernels given to each model was varied so that the number of parameters in each model configuration was equal to each other and the hGRU (36, 16, and 9 kernels). We also tested two other feedforward models that featured candidate operations for incorporating contextual information into local convolutional activities. One version used (2-pixel) dilated convolutions, which involves applying a stride to the kernel before convolving the input [44, 45], and has been found useful for many computer vision problems [38, 46, 47]. The other version applied a non-local operation to convolutional activities [48], which can introduce (non-recurrent) interactions between units in a layer. These operations were incorporated into the first feature extraction layer of the medium kernel ($15 \times 15$ filter) model described above. We also considered deeper versions of each of the above "1-layer" models (referring to the depth of the feature extraction stage), stacking them to build 3- and 5-layer versions. This yielded a total of 15 different feedforward models.

Without exception, the performance of each feedforward model was significantly strained by the Pathfinder challenge. The magnitude of this straining was well predicted by model depth and size, and operations for incorporating additional contextual information made no discernible difference to the overall pattern of results. The 1-layer models were most effective on the 6-length Pathfinder dataset, but were unable to do better than chance on the remaining conditions. Increasing model capacity to 3 layers rescued the performance of all but the small kernel model on the 9-length Pathfinder dataset,

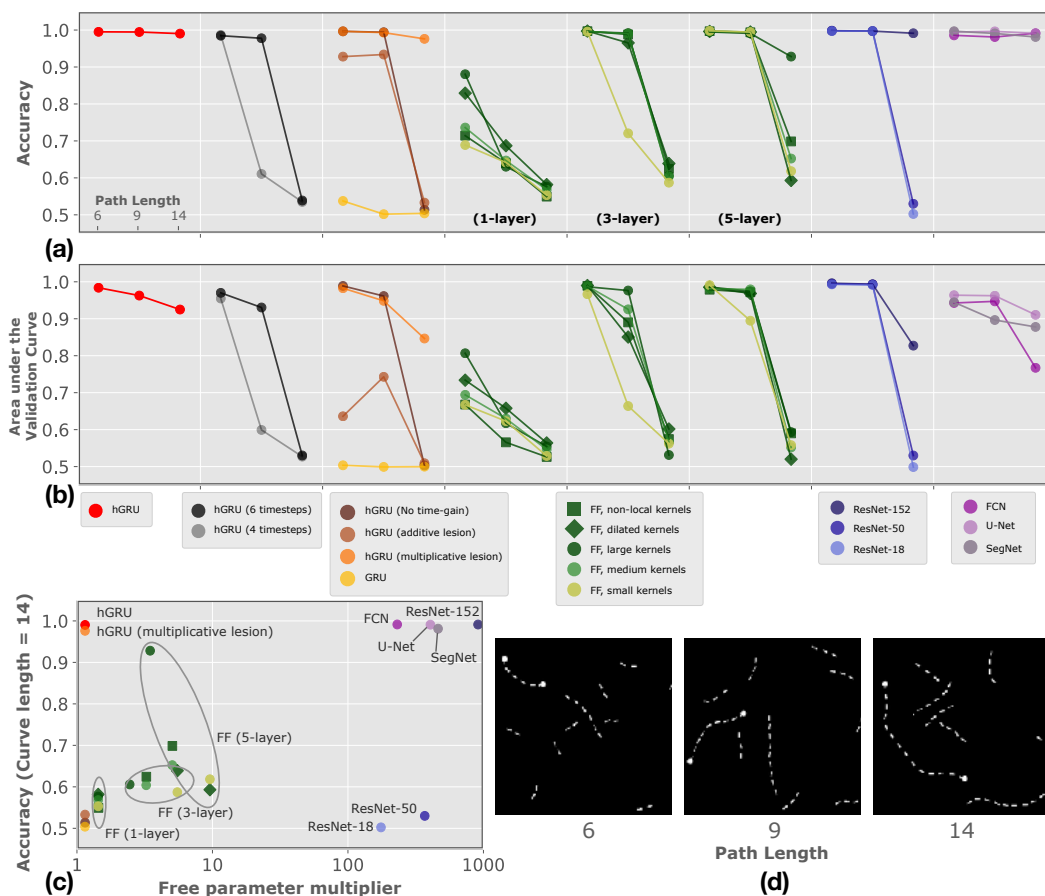

Figure 3: The hGRU efficiently learns long-range spatial dependencies that otherwise strain feedforward architectures. **(a)** Model accuracy is plotted for the three Pathfinder challenge datasets, which featured paths of 6- 9- and 14-paddles. Each panel depicts the accuracy of a different model class after training on for each pathfinder dataset (see Supp. Material for additional models). Only the hGRU and state-of-the-art models for classification (the two right-most panels) approached perfect accuracy on each dataset. **(b)** Measuring the area under the learning curve (ALC) of each model (mean accuracy) demonstrates that the rate of learning achieved by the hGRU across the Pathfinder challenge is only rivaled by the U-Net architecture (far right). **(c)** The hGRU is significantly more parameter-efficient than feedforward models at the Pathfinder challenge, with its closest competitors needing at least $200\times$ the number of parameters to match its performance. The x-axis shows the number of parameters in each model versus the hGRU, as a multiple of the latter. The y-axis depicts model accuracy on the 14-length Pathfinder dataset. **(d)** Pathfinder challenge exemplars of different path lengths (all are positive examples).

but even then did little to improve performance on the 14-length dataset. Of the 5-layer models, only the large kernel configuration came close to solving the 14-length dataset. The ALC of this model, however, demonstrates that its rate of learning was slow, especially compared to the hGRU (Fig. 3b). The failures of these feedforward models is all the more striking when considering that each had between $1\times$ and $10\times$ the number of parameters as the hGRU (Fig. 3c, compare the red and green markers).

**Residual networks** We reasoned that if the performance of feedforward models on the Pathfinder challenge is a function of model depth, then state-of-the-art networks for object recognition with many times the number of layers should easily solve the challenge. We tested this possibility by

training ResNets with 18, 50, and 152 layers on the Pathfinder challenge. Each model was trained "from scratch" with standard weight initialization [49], and given additional epochs of training (4) to learn the task because of their large number of parameters. However, even with this additional training, only the deepest 152-layer ResNet was able to solve the challenge (Fig. 3a). Even so, the 152-layer ResNet was less efficient at learning the 14-length dataset than the hGRU (Fig. 3b), and achieved its performance with nearly $1000\times$ as many parameters (Fig. 3c; see Supp. Material for additional ResNet experiments).

**Per-pixel prediction models**    We considered the possibility that CNN architectures for per-pixel prediction tasks, such as contour detection and segmentation, might be better suited to the Pathfinder challenge than those designed for classification. We therefore tested three representative per-pixel prediction models: the fully-convolutional network (FCN), the skip-connection U-Net, and the unpooling SegNet. These models used an encoder/decoder style architecture, which was followed by the readout processing stage of the standard architecture described above to make them suitable for Pathfinder classification. Encoders were the VGG16 [50], and each model was trained from scratch with Xavier initialized weights. Like the ResNets, these models were given 4 epochs of training to accommodate their large number of parameters.

The fully-convolutional network (FCN) architecture is one of the first successful uses of CNNs for per-pixel prediction [3, 44, 51, 52]. Decoders in these models use "$1\times1$" convolutions to combine upsampled activity maps from several layers of the encoder. We created an FCN model which applied this procedure to the last layer of each of the 5 VGG16-convolution blocks. These activity maps were upsampled by learnable kernels, which were initialized with weights for bilinear interpolation. In contrast to the feedforward models discussed above, the FCN successfully learned all conditions in the Pathfinder challenge (Fig. 3a, purple circle). It did so less efficiently than the hGRU, however, with a lower ALC score on the 14-length dataset and $200\times$ as many free parameters (Fig. 3b).

Another approach to per-pixel prediction uses "skip connections" to connect specific layers of a model's encoder to its decoder. This approach was first described in [44] as a method for more effectively merging coarse-layer information into a model's decoder, and later extended to the U-Net [53]. We implemented a version of the U-Net architecture that had a VGG16 encoder and a decoder. The decoder consisted of 5 randomly initialized and learned upsampling layers, which had additive connections to the final convolutional layer in each of the encoder's VGG16 blocks. Using standard VGG16 nomenclature to define one of these connections, this meant that "conv 4_3" activity from the encoder was added to the second upsampled activity map in the decoder. The U-Net was on par with the hGRU and the FCN at solving the Pathfinder challenge. It was also nearly as efficient as the hGRU in doing so (Fig. 3b), but used over $350\times$ as many parameters as the hGRU (Fig. 3c; see Supp. Materials for additional U-Net experiments).

Unpooling models eliminate the need for feature map upsampling by routing decoded activities to the locations of the winning max-pooling units derived from the encoder. Unpooling is also a leading approach for a variety of dense per-pixel prediction tasks, including segmentation, which is exemplified by SegNet [54]. We tested a SegNet on the Pathfinder challenge. This model has a decoder that mirrors its encoder, with unpooling operations replacing its pooling. The SegNet achieved high accuracy on each of the Pathfinder datasets, but was less efficient at learning them than the hGRU, with worse ALC scores across the challenge (Fig. 3b). The SegNet also featured the second-most parameters of any model tested, which was $400\times$ more than the hGRU.

## 4    Explaining biological horizontal connections with the hGRU

Statistical image analysis studies have suggested that cortical patterns of horizontal connections, commonly referred to as "association fields", may reflect the geometric regularities of oriented elements present in natural scenes [55]. Because the hGRU is designed to capture such spatial regularities, we investigated whether it learned patterns of horizontal connections that resemble these association fields. We visualized the horizontal kernels learned by the hGRU to solve tasks (Fig. S5). When trained on the the Pathfinder challenge, hGRU kernels resembled the dominant patterns of horizontal connectivity in visual cortex. Prominent among these patterns are (1) the antagonistic near-excitatory vs. far-inhibitory surround organization also found in the visual cortex [56]; (2) the association field, with collinear excitation and orthogonal inhibition [8, 9]; and (3) other higher-order surround computations [57]. We also visualized these patterns after training the hGRU to detect

contours in the naturalistic BSDS500 image dataset [1]. These horizontal kernels took on similar patterns of connectivity, but with far more definition and regularity, suggesting that the hGRU learns best from natural scene statistics.

How well does the hGRU explain human psychophysics data? We tested this by recreating the synthetic contour detection dataset used in [22]. This task had human participants detect a contour formed by co-linearly aligned paddles in an array of randomly oriented distractors. Multiple versions of the task were created by varying the distance between paddles in the contour (5 conditions). Contour detection accuracy of the hGRU was recorded on each of dataset for comparison with participants in [22], whose responses were digitally extracted with WebPlotDigitizer from [22] and averaged (N=2). Plotting hGRU accuracy against the reported "detection score" revealed that increasing inter-paddle distance caused similar performance straining for both (Fig. S6).

## 5 Discussion

The present study demonstrates that long-range spatial dependencies generally strain CNNs, with only very deep and state-of-the-art networks overcoming the visual variability introduced by long paths in the Pathfinder challenge. Although feedforward networks are generally effective at learning and detecting relatively rigid objects shown in well-defined poses, these models tend towards a brute-force solution when tasked with the recognition of less constrained structures, such as a path connecting two distant locations. This study adds to a body of work highlighting examples of routine visual tasks where CNNs fall short of human performance [58–62].

We demonstrate a solution to the Pathfinder challenge inspired by neuroscience. The hGRU leverages computational principles of visual cortical circuits to learn complex spatial interactions between units. For the Pathfinder challenge, this translates into an ability to represent the elements forming an extended path while ignoring surrounding clutter. We find that the hGRU can reliably detect paths of any tested length or form using just a single layer. This contrasts sharply with the successful state-of-the-art feedforward alternatives, which used much deeper architectures and orders of magnitude more parameters to achieve similar success. The key mechanisms underlying the hGRU's performance are well known in computational neuroscience [10, 16–20]. However, these mechanisms have been typically overlooked in computer vision (but see [39] for a successful vision model using horizontal connections and shown to break text-based CAPTCHAs).

We also found that hGRU performance on the Pathfinder challenge is a function of the *amount of time* it was given for processing. This finding suggests that it concurrently expands the facilitative influence of one end of a target curve to the other while suppressing the influence of distractors. The performance of the hGRU on the Pathfinder challenge captures the iterative nature of computations used by our visual system during similar tasks [63] – exhibiting a comparable tradeoff between performance and processing-time [7]. Visual cortex is replete with association fields that are thought to underlie perceptual grouping [11, 64]. Theoretical models suggest that patterns of horizontal connections reflect the statistics of natural scenes, and here too we find that horizontal kernels in the hGRU learned from natural scenes resemble cortical patterns of horizontal connectivity, including association fields and the paired near-excitatory / far-inhibitory surrounds that may be responsible for many contextual illusions [20, 56]. The horizontal connections learned by the hGRU reproduce another aspect of human behavior, in which the saliency of a straight contour decreases as the distance between its paddles increases. This sheds light on a possible relationship between horizontal connections and saliency computation.

In summary, this work diagnoses a computational deficiency of feedforward networks, and introduces a biologically-inspired solution that can be easily incorporated into existing deep learning architectures. The weights and patterns of behavior learned by the hGRU appear consistent with those associated with the visual cortex, demonstrating its potential for establishing novel connections between machine learning, cognitive science, and neuroscience.

### Acknowledgments

This research was supported by NSF early career award [grant number IIS-1252951] and DARPA young faculty award [grant number YFA N66001-14-1-4037]. Additional support was provided by the Carney Institute for Brain Science and the Center for Computation and Visualization (CCV) at Brown University.

## Footnotes

[1] There are four separate connectivity patterns in [20] to describe inhibition vs. excitation and near vs. far interactions between units. We combine these into a separate inhibitory vs. excitatory kernels to simplify notation.

[2] These modifications involved relaxing several constraints from the original neuroscience model that are less useful for solving the tasks investigated here (see Supp. Material for performance of an hGRU with constrained inhibition and excitation.)

[3]GRU gate activities are often a function of a hidden state and $\mathbf{X}[t]$. Because the feedforward drive here is constant w.r.t. time, we omit it from these calculations. In practice, its inclusion did not affect performance.

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
