[Supplementary Material]

# Supplementary Material for: Learning long-range spatial dependencies with horizontal gated recurrent units

**Drew Linsley, Junkyung Kim, Vijay Veerabadran, Charlie Windolf, Thomas Serre**
Department of Cognitive Linguistic & Psychological Sciences
Carney Institute for Brain Science
Brown University
Providence, RI 02912
{drew_linsley,junkyung_kim, vijay_veerabadran,
charles_windolf, thomas_serre}@brown.edu

## Deriving the hGRU from the contextual neural circuit model

From Eq. 1, we obtain the following by rearranging the decay term:

$$\dot{H}_{xyk}^{(1)} = -\eta^{-1}\epsilon^2 H_{xyk}^{(1)} + \eta^{-1}\left[\xi X_{xyk} - (\alpha H_{xyk}^{(1)} + \mu)C_{xyk}^{(1)}\right]_+$$

$$\dot{H}_{xyk}^{(2)} = -\eta^{-1}\sigma^2 H_{xyk}^{(2)} + \tau^{-1}\left[\gamma C_{xyk}^{(2)}\right]_+, \tag{1}$$

where $[\cdot]_+ = \max(\cdot, 0)$ is the ReLU function.

Now, we would like to discretize this equation using Euler's method. We first simplify the above equation by choosing $\eta = \tau$ and $\sigma = \epsilon$:

$$\dot{H}_{xyk}^{(1)} = -\eta^{-1}\epsilon^2 H_{xyk}^{(1)} + \eta^{-1}\left[\xi X_{xyk} - (\alpha H_{xyk}^{(1)} + \mu)C_{xyk}^{(1)}\right]_+$$

$$\dot{H}_{xyk}^{(2)} = -\eta^{-1}\epsilon^2 H_{xyk}^{(2)} + \eta^{-1}\left[\gamma C_{xyk}^{(2)}\right]_+,$$

Now, apply Euler's method with timestep $h$. This gives the following difference equation:

$$H_{xyk}^{(1)}[t] = H_{xyk}^{(1)}[t-1] + h\left(-\eta^{-1}\epsilon^2 H_{xyk}^{(1)}[t-1] + \eta^{-1}\left[\xi X_{xyk}[t-1] - (\alpha H_{xyk}^{(1)}[t-1] + \mu)C_{xyk}^{(1)}[t]\right]_+\right)$$

$$H_{xyk}^{(2)}[t] = H_{xyk}^{(2)}[t-1] + h\left(-\eta^{-1}\epsilon^2 H_{xyk}^{(2)}[t-1] + \eta^{-1}\left[\gamma C_{xyk}^{(2)}[t]\right]_+\right).$$

Distributing $h$, we get

$$H_{xyk}^{(1)}[t] = H_{xyk}^{(1)}[t-1] - h\frac{\epsilon^2}{\eta}H_{xyk}^{(1)}[t-1] + h\eta^{-1}\left[\xi X_{xyk}[t-1] - (\alpha H_{xyk}^{(1)}[t-1] + \mu)C_{xyk}^{(1)}[t]\right]_+$$

$$H_{xyk}^{(2)}[t] = H_{xyk}^{(2)}[t-1] - h\frac{\epsilon^2}{\eta}H_{xyk}^{(2)}[t-1] + h\eta^{-1}\left[\gamma C_{xyk}^{(2)}[t]\right]_+.$$

Now, notice that if we choose $h = \frac{\eta}{\epsilon^2}$, the first two terms on the RHS of each line will cancel:

$$H_{xyk}^{(1)}[t] = \epsilon^{-2}\left[\xi X_{xyk}[t-1] - (\alpha H_{xyk}^{(1)}[t-1] + \mu)C_{xyk}^{(1)}[t]\right]_+$$

$$H_{xyk}^{(2)}[t] = \epsilon^{-2}\left[\gamma C_{xyk}^{(2)}[t]\right]_+. \tag{2}$$

Figure S1: Circuit diagrams of the gated recurrent unit (GRU) and the hGRU showing the exitatory (red) and inhibitory (blue) flow of information. (a) A reference diagram of the GRU, demonstrating how the kernels $U^{(1)}$ and $U^{(2)}$ act as the "gain" and "mix" on the circuit's computations: selecting feature channels from $H[t-1]$ to process on the current timestep, and determining how to mix a candidate $\tilde{H}[t]$ with $H[t-1]$ to produce $H[t]$. Candidate activities are calculated by applying a hyperbolic tangent (tanh) to a linear combination of the feedforward drive $X$ and the convolution of the kernel $W$ with the new "gain"-modulated hidden state. (b) The same gain and mix are used in the hGRU in a slightly different configuration. Note that in the hGRU, the mix is a function of $\mathbf{H}^{(1)}[t]$ instead of $\mathbf{H}^{(2)}[t-1]$. The separate processing stages of the hGRU, inspired by the recurrent neural circuit of [11], are critical for learning the Pathfinder challenge.

We now have a discrete-time approximation of the initial dynamical system, shown in Eq. 2 in the main text. Note that this model can also be thought of as a convolutional RNN with ReLU nonlinearity. Because RNNs are difficult to train in practice, the state of the art is to incorporate learned "gates" to manage the flow of information over time [1].

$$
\begin{aligned}
G_{xyk}^{(1)}[t] &= \sigma\left(X_{xyk} + (\mathbf{U}^{(1)} * \mathbf{H}^{(2)}[n-1])_{xyk} + b_k^{(1)}\right) \\
G_{xyk}^{(1)}[t] &= \sigma\left(X_{xyk} + (\mathbf{U}^{(2)} * \mathbf{H}^{(1)}[n])_{xyk} + b_k^{(2)}\right),
\end{aligned}
\tag{3}
$$

where $\sigma$ is a squashing pointwise nonlinearity, $\mathbf{U}^{(\cdot)}$ are convolution kernels, and $b^{(\cdot)}$ are bias vectors. Applied to Eq. 2, these gates integrate past information with new computations more flexibility than the Euler integration above:

$$
\begin{aligned}
H_{xyk}^{(1)}[t] &= \epsilon^{-2} G_{xyk}^{(1)}[t]\left[\xi X_{xyk}[t-1] - (\alpha H_{xyk}^{(1)}[t-1] + \mu)C_{xyk}^{(1)}[t]\right]_+ \\
&\quad + (1 - G_{xyk}^{(1)}[t])H_{xyk}^{(1)}[t-1]
\end{aligned}
\tag{4}
$$

$$
H_{xyk}^{(2)}[t] = \epsilon^{-2} G_{xyk}^{(2)}[t]\left[\gamma C_{xyk}^{(2)}[t]\right]_+ + (1 - G_{xyk}^{(2)})H_{xyk}^{(2)}[t-1]
$$

This defines the simplest gated version of the RNN described in Eq. 2. As discussed in the main text, we modified the model in several other ways to improve its effectiveness on the visual tasks examined here: (1) We found that using the GRU-style "gain" (eq 3.) to control the first stage of horizontal interactions was more effective than using "mixing" gates in both stages. (2) We found it effective to replace the $\epsilon^{-2}$ scaling by the learned per-timestep parameter $\boldsymbol{\eta}$, which can be thought of as a restricted application of batch normalization to the RNN's hidden activity to control saturating activities stabilize training [2]. (3) We included both linear and quadratic forms of excitation in $\mathbf{H}^{(2)}$ for symmetry with $\mathbf{H}^{(1)}$. Enabling the model to spread excitation via both linear and quadratic propagation is potentially useful for propagating activity based on either first- (e.g., contrast strength) or second-order (e.g., self-similarity) statistical regularities. The linear and quadratic terms are scaled per-feature by the learnable parameters $\boldsymbol{\alpha}$ and $\boldsymbol{\mu}$ when calculating $\mathbf{H}^{(1)}$, and $\boldsymbol{\kappa}$ and $\boldsymbol{\omega}$, and $\boldsymbol{\omega}$ in $\mathbf{H}^{(2)}$. See S1 for a comparison of circuit diagrams for the GRU vs. the hGRU.

## The Pathfinder challenge

**Overview**    The main goal of the Pathfinder challenge is to assess the ability of a computer vision system to determine whether two distant locations in an image, marked by while circles, are connected

Figure S2: Image generator algorithm. (a) Depiction of image generation shown in seven steps. (b) Depiction of the geometric constraints for trailing paddle placement during path generation.

by a path. The stimulus dataset consists of binary images, each containing two white circles, two white 'target' paths, and multiple distractor paths, placed on a black background. All images used in our experiments are rendered on a 150×150 canvas. The task is inspired by [3].

Each image is generated by first sampling two target paths and then rendering multiple distractor paths. Each path consists of $k$ co-circularly arranged identically shaped "paddles" of length $l$ and thickness $d$ (Fig. S2b). A paddle is a white oriented bar, characterized by the position of its center and its orientation, $\theta$. Each path is generated inductively, namely, by first sampling the position of its "seed paddle" which serves as the end of the path and then iteratively adding new "trailing paddles" next to the seed paddle or the last trailing paddle. The high-level overview of the image generation algorithm is depicted in Fig. S2 and the list of image parameters in Table S2.

**Positioning target paths**  The first stage of generating target paths involves randomly sampling the position of an invisible circle of radius $r$ (Fig. S2a, step 1) within an empty image. Then, a randomly oriented line pivoted at the center of the circle makes two intersections with the circle. These intersections serve as the positions of two target paths' seed paddles (Fig. S2a, step 2). This constraint ensures that target paths are always located within a sufficient proximity to each other. Without this constraint, target paths can be separated by an arbitrary distance, which may allow a model to make classification decision solely based on the distance between the white markers, for the markers will tend to be located much farther apart in negative examples. Then, two randomly oriented paddles are rendered at each of the intersections (Fig. S2a, step 3), serving as seed paddles of the target paths.

**Growing a path**  Trailing paddles are sequentially added to the end of each target path (Fig. S2a, step 4 and 5). Each trailing paddle is added by first sampling its orientation, $\theta_i$, according to the probability distribution defined by the angle formed between the new and the previous trailing paddle, $\Delta\theta = \theta_i - \theta_{i-1}$:

$$P(\theta_i) = \frac{1}{Z} \max(\cos{(c\Delta\theta)}, 0) \tag{5}$$

$$\text{where } Z = \int_{-\frac{\pi}{2}}^{\frac{\pi}{2}} \cos(\theta)d\theta = 2 \tag{6}$$

Note that the continuity parameter $c$ determines the overall rigidity of a path by constraining the possible range of orientations of each new trailing paddle. With the sampled orientation, the new trailing paddle's position is determined such that the line extending from the trailing paddle intersects with the line extending from the new paddle $m$ pixels away from the end of the two paddles while also forming an angle $\Delta\theta$ (Fig. S2b). Note that the parameter $m$ determines the margin between adjacent paddles in a path. Because we do not allow paddles to cross or touch, we add new paddles to the two target paths in alternation to ensure that the shapes of one path does not restrict the shape of the other. The total number of target path paddles, or simply length of target paths, is denoted by the parameter $k$.

Table 1: Image parameters in the Pathfinder challenge.

| Notation | Definition |
| --- | --- |
| $r$ | Radius of a circle |
| $k$ | Target path length, or the number of paddles in a target path |
| $l$ | Paddle length, in pixels |
| $d$ | Paddle thickness, in pixels |
| $m$ | Inter-paddle margin, in pixels |
| $c$ | Continuity |

Additional distractor paths consisting of $\frac{k}{3}$ paddles are added to the image. Their generation process is identical to that of target paths, except that a seed paddle's location is independently sampled. The total number of distractor paths is chosen so that the total number of paddles in an image is 150. The usage of two target paths as well as distractor paths prevents any 'short-cut' solution to this problem, such as detecting a loose end of a path or a lone white circle as a local diagnostic cue for classification. Additionally, this design allows us to ensure that positive and negative images are indistinguishable in terms of the way paddles are arranged. This forces a model to perform classification solely based on the connectedness of the white circles.

As mentioned, the generator does not allow paddles to be too close to each other or make any contact. If a newly sampled paddle is making contact with or not separated from other paddles by at least $m$ pixels, the generator rejects it and re-samples a paddle. In practice, this is done by applying circular dilation on the paddles using a kernel of radius $m$ and checking if the dilated images of paddles has any overlap.

**Shape variability**  Because adding an additional paddle to a path multiplicatively increases the number of possible shapes of the path, path shape variability is exponential to the total number of paddles in a path, $k$. Similarly, the continuity parameter $c$ controls path shape variability in an exponential manner because it scales the range of possible angles formed between every pair of adjacent paddles in a path. In our experiment, we vary path length $k$ to examine how model performance changes as the shape variability of a path in a dataset increases. All images in the Pathfinder challenge are generated using Python and OpenCV.

## The Synthetic Contours Dataset

In this section, we shall explain the motivation and construction procedure for the Synthetic Contours Dataset. We created the Synthetic Contours Dataset (SCD) to compare an hGRU pre-trained on the BSDS500 [4] for contour detection against human behavioral data as described in [5].

**Dataset generation**  We generated the SCD by following closely, and extending the dataset construction procedure described in [5]. SCD enables us to parametrically control the position, orientation, and relative spacing of generated contours, resulting in 28 million images with practically infinite variability along these directions. All images in the SCD have a resolution of 256×256 pixels, spanning a radius of 4 visual degrees. Each contour image is rendered by placing "paddles" of length 0.1 visual degrees within uniformly spaced cells on a master grid. This master grid covers the entire image with a height and a width of 8 visual degrees. Each cell in the grid spans a height and width of 0.25 visual degrees. A total of 32 paddles are placed along every row/column of the master grid.

A contour of length $l$ is generated in the master grid by filling $l$ neighboring grid cells on any grid-diagonal, and each paddle connects the diagonally opposite points in their respective cells. This aligns the paddles in a collinear manner to form a contour of length $l$. In order to ensure that the majority of the contour stays within the image, its center is positioned within a maximum distance from the center of the image. This distance from the image center, known as eccentricity, is denoted as $e$. Once the contour is generated, all remaining unoccupied cells are filled with paddles each with an orientation $\theta \in [0, 2\pi]$ sampled from a uniform distribution.

**Parametric dataset variability**  Contours in SCD are formed by regulating the dataset variability in 2 different directions: (1) contour length and (2) relative spacing. We denote the number of paddles present in a contour as its length. Each image in the SCD contains a contour with a length of either

5, 9, 14 or 17 paddles. The distance between neighboring paddles that form a contour is denoted as the relative spacing. While we vary the relative spacing, the global spacing between distractor paddles is maintained constant. Following [5], we apply a shear operation to the generated image by a shear factor *SF*, which controls how close or far the neighboring contour paddles are positioned. We generated images with 5 different relative-spacing conditions by varying SF between -0.6 and +0.6. A comparison between human performance and model performance on the task of contour detection can be found in Fig. S6. Example figures with different combinations of contour length and relative spacing are demonstrated in Fig. S3. All the images in SCD were generated using the Python Psychophysics toolbox, Psychopy [6].

Figure S3: Examples of images from the SCD. Unlike in the dataset, contours in the figure are highlighted with a different color for the reader's convenience. (a) Contour with *SF*=-0.6, *l*=17, closest relative spacing. (b) Contour with *SF*=-0.4, *l*=14. (c) Contour with *SF*=0.2, *l*=9. (d) Contour with *SF*=0.6, *l*=9, farthest relative spacing.

## Model architecture details

All of the recurrent models and feedforward models (shown in shades of green in Fig. 3a in the main text) tested in our study share the 'input stage' as the preprocessing module and the 'readout stage' as the classification module, which allows us to more directly compare the effectiveness of different architectures (called 'feature extraction' stage) at solving the Pathfinder challenge. Unless otherwise specified, all feedforward model weights were Xavier initialized [7]. Recurrent model kernels and parameters were also initialized with Xavier, while their biases were chronos-initialized [1]. Hidden states of te recurrent models were also randomly initialized. Implementations of the hGRU can be found at `https://github.com/serre-lab/hgru_share`, and datasets are available upon request.

Standard cross entropy was used to compute loss. Models were trained with gradient descent using the Adaptive Moment Estimation (Adam) optimizer [8] with base learning rate of $10^{-3}$ on batches of 32 images. Our experiments are conducted using TensorFlow [9].

**Input stage**  The input stage consists of a convolutional layer with 25 kernels of size 7×7. The kernels are initialized as Gabor filters at 12 orientations and 2 phases, plus a radially symmetric difference-of-Gaussian filter. Filter activity undergoes a pointwise squaring non-linearity before being passed to the model-specific feature extraction stage. Note that none of the off-the-shelf models (residual networks and per-pixel prediction models) use this input stage.

**Readout stage**  The readout stage takes the output from the final layer of the feature extraction stage and computes a two-dimensional likelihood vector for deriving a decision on an image from the Pathfinder challenge. The readout stage contained three parts: (1) a 1x1 convolutional filter that transforms the multi-channel output map from the feature extraction stage to a two-channel map, each corresponding to the positive/negative class. (2) A batch-normalized [10] global max pool, whose output represents in each channel the maximum activation value across space. (3) A linear classifier which maps the two-dimensional feature vector into a class decision. The configuration of this readout stage allows us to compare the accuracy between models designed for per-image prediction with those designed for per-pixel prediction. Note that residual networks do not use a readout stage as they are proposed as standalone image classification architectures.

**Additional recurrent models** We tested three other kinds of recurrent models that were not discussed in the main text. These were the hGRU with batchnorm (hGRU-BN), the hGRU with a nonnegative pointwise nonlinearty (hGRU-ReLU), and a two-layer GRU (GRU-2L).

The hGRU formulation in the main text incorporates a highly constrained form of the normalization discussed in [2], in which a learned scalar modulates activity at every timestep to help control unstable gradients. We found that incorporating the full batchnorm formulation of [2] in the hGRU also performed well on the Pathfinder challenge. We implemented the resulting hGRU-BN by adding batchnorm to multiple computations in the circuit. While it is possible to share batchnorm parameters across timesteps, the model performs better when they are not shared. In this formulation, $H^{(1)}[t]_{xyk}$ is calculated as

$$
\begin{aligned}
\mathbf{G}^{(1)}[t] &= \sigma(\mathrm{BN}(\mathbf{U}^{(1)} * \mathbf{H}^{(2)}[t-1])) \\
C^{(1)}[t]_{xyk} &= \mathrm{BN}(\mathbf{W} * (\mathbf{G}^{(1)}[t] \odot \mathbf{H}^{(2)}[t-1]))_{xyk} \\
H^{(1)}_{xyk}[t] &= \zeta(X_{xyk} - C^{(1)}_{xyk}[t](\alpha_k H^{(2)}_{xyk}[t-1] + \mu_k))
\end{aligned}
\tag{7}
$$

where

$$
\mathrm{BN}(\mathbf{h}; \boldsymbol{\delta}, \boldsymbol{\nu}) = \boldsymbol{\nu} + \boldsymbol{\delta} \odot \frac{\mathbf{h} - \widehat{\mathbb{E}}[\mathbf{h}]}{\sqrt{\widehat{\mathrm{Var}}[\mathbf{h}] + \epsilon}}
$$

Where $\mathbf{h} \in \mathbb{R}^d$ is the vector of preactivations being normalized by batchnorm and $\boldsymbol{\delta}, \boldsymbol{\nu} \in \mathbb{R}^d$ are parameters that control the standard deviation and mean of the normalized activities, $\epsilon$ is a regularization hyperparameter, and $\odot$ is elementwise multiplication. As in the hGRU described in the main text, this updated $\mathbf{H}^{(1)}[t]$ is next used to calculate $\mathbf{H}^{(2)}[t]$.

$$
\begin{aligned}
\mathbf{G}^{(2)}[t] &= \sigma(\mathrm{BN}(\mathbf{U}^{(2)} * \mathbf{H}^{(1)}[t]))) \\
C^{(2)}[t]_{xyk} &= \mathrm{BN}(\mathbf{W} * \mathbf{H}^{(1)}[t])_{xyk} \\
\tilde{H}^{(2)}_{xyk}[t] &= \zeta(\kappa_k H^{(1)}_{xyk}[t] + \beta_k C^{(2)}_{xyk}[t] + \omega_k H^{(1)}_{xyk}[t] C^{(2)}_{xyk}[t]) \\
H^{(2)}_{xyk}[t] &= H^{(2)}_{xyk}[t-1](1 - G^{(2)}_{xyk}[t]) + \tilde{H}^{(2)}_{xyk}[t] G^{(2)}_{xyk}[t]
\end{aligned}
\tag{8}
$$

Performance of the hGRU-BN can be found in S4 as "hGRU (batchnorm)". In addition to its ability to solve the Pathfinder challenge, the hGRU-BN is far more stable during training than the original hGRU formulation when its pointwise nonlinearity $\zeta$ is substituted for nonlinearities that are not squashing. This allowed us to build a version of the hGRU where the pointwise nonlinearity $\zeta$ is set to linear rectification (ReLU), which constrains $\mathbf{H}^{(1)}$ to inhibition and $\mathbf{H}^{(2)}$ to excitation as is done in the original model by [11]. Also similar to [11] this model used separate kernels $\mathbf{W}^I$ and $\mathbf{W}^E$ to calculate facilitative vs. suppressive horizontal interactions at $\mathbf{H}^{(1)}$ and $\mathbf{H}^{(2)}$. This model solved the pathfinder challenge, and is denoted in Fig. S4 as "hGRU (nonnegative)".

We also tested whether the hGRU's ability to solve the Pathfinder challenge was merely a function of its two processing stages. We developed a GRU with two layers of convolution (with separate kernels) to calculate its candidate hidden state. Although this "GRU (2L)" model performed better than the typical one-layer version on the 6-length pathfinder, it was unable to solve the other versions of the problem (Fig. S4).

**Additional models** We extended our screening of feedforward architectures on the Pathfinder challenge to include three new model classes. The first combines highway network modules [12] with the input- and readout-stages that were used to screen feedforward model operations in the main text. As with those models, we tested the highway network in one-, three-, and five-layer configurations. The highway network models performed similarly to the "large-kernel" configuration of feedforward models screened in the main text, solving the 6- and 9-length pathfinder tasks, but straining on the 14-length one (see "Hwy-Net" in Fig. S4).

We also tested "constrained" versions of our top-performing feedforward and per-pixel prediction models, the ResNet-152 and the U-Net. We created new versions of these models that had $\frac{1}{2}$, $\frac{1}{5}th$,

Figure S4: A comparison of the hGRU and additional control models on the Pathfinder challenge. These models are hGRUs with additional normalization stages ("batchnorm") or constraints ("non-negative"); a GRU with two layers of processing; feedforward models with highway network layers; 152-layer ResNets restricted to a fraction of the total parameters of the original model; and U-Net models restricted to a fraction of the total parameters of the original model. (a) While all versions of the hGRU have high accuracy on all three Pathfinder challenge datasets, only the U-Net limited to $\frac{1}{2}$ or $\frac{1}{5}$ the total number parameters performs similarly. (b) Model accuracy is plotted as a function of the multiple of parameters each has the hGRU (i.e., hGRU is 1 and all models with more parameters are $> 1$). This reveals a model complexity and performance gap between the hGRU and other classes of feedforward models. Per-pixel prediction models, such as the U-Net, perform relatively well on the Pathfinder challenge but need many more parameters than the hGRU to do so: note the difference in performance between the U-Net ($\frac{1}{10}$) and the hGRU. Highway nets seem to follow a similar performance trajectory, and 152-layer ResNets struggle with the task without their full set of parameters (see Fig. 3 in the main text).

or $\frac{1}{10}th$ the total number of parameters. The resulting ResNet models solved the 6- and 9-length pathfinder datasets were unsuccessful on the 14-length pathfinder. In constrast, the constrained U-Nets performed far better, with only the U-Net $\frac{1}{10}$ strained by the 14-length pathfinder. Plotting model performance as a function of their multiple of parameters vs. the hGRU reveals a performance shelf for these feedforward models that depend on depth to expand their receptive fields and solve tasks such as the Pathfinder challenge. This strategy is less efficient than the hGRU's ability to form interactions between units *at a single processing layer*. The model with the closest performance and number of parameters as the hGRU in this comparison is the $\frac{1}{5}$ U-Net, which still needed over an order of magnitude more free parameters to solve the challenge.

## Training on natural images and comparisons to human data

**Visualizing horizontal connections**   Theoretical models of visual cortex suggest that patterns of horizontal connections reflect statistical regularities of oriented features in natural scenes [13]. Here too we find that horizontal kernels in the hGRU learned from both synthetic images from the Pathfinder challenge and natural scenes resemble cortical patterns of horizontal connectivity, including association fields and the paired near-excitatory and far-inhibitory surrounds (Fig. b).

We visualized hGRU horizontal connectivity in models through a two-step procedure that aligned kernels into a common reference orientation (90 degrees), and then used PCA to identify and denoise common patterns. For each input channel of the hGRU's horizontal kernel, we rotated the kernels in the opposite orientation of the previous layer's filter used to compute their feedforward drive. For example, for hGRUs trained on the Pathfinder challenge, the feedforward drive came from kernels initialized with oriented gabors. Here, a gabor oriented around 90 degrees would result in its horizontal kernels being rotated -90 degrees. After normalizing the orientation of all horizontal kernels, they were mean centered and then passed through PCA to visualize their "eigenconnectivity" patterns in Fig. . In other words, these eigenconnectivities correspond to the spatial associations learned by horizontal kernels, aggregated over all orientation channels. Eigenconnectivity patterns are sorted by their eigenvalues (80% cumulative variance cutoff).

**(a)**

**(b)**

Figure S5: hGRU eigenconnectivity on when trained on the Pathfinder challenge and contour detection in natural scenes. (a) Eigenconnectivity of hGRU models trained on 6-, 9-, and 14-length Pathfinder datasets. (b) Eigenconnectivity of an hGRU on the BSDS500 contour detection dataset. The exhibited kernels contain patterns that are similar to those observed in visual cortex.

We plot hGRU eigenconnectivities learned from two tasks: the Pathfinder challenge (Fig. a) and contour detection in natural images (Fig. b). For the first task, we use models described in the main text trained on each of the Pathfinder datasets. For the second task, we trained a new version of the hGRU to detect contours in the BSDS500 [4]. We used an 8-timestep with the first two blocks of PASCAL weight initialized convolutional layers from the VGG16 as its input processing stage. This model was trained for 1000 epochs on the 200 training images in the BSDS500, using data augmentations (random crops to 300×300 pixels, random rotations of +/- 30°, and random up/down/left/right flips), a per-pixel cross-entropy loss, and the same learning rate and optimizer as the models described above.

Pathfinder challenge eigenconnectivity is similar to canonical patterns of cortex, including the (1) the antagonistic near-excitatory vs. far-inhibitory surround organization of hypercolumns in the visual cortex [14]; (2) the association field, with collinear excitation and orthogonal inhibition [15, 16]; and (3) higher-order surround computations [17].

Horizontal kernel eigenconnectivities are far cleaner and qualitatively more similar to purported canonical patterns of connectivity in cortex after training the hGRU to detect contours in natural images (compare Fig. a and Fig. b). We believe that such a difference at least in part results from the the kinds of image statistics that are useful for the respective tasks of Pathfinder vs. BSDS – path integration vs. contour detection. The path integration task in the Pathfinder challenge can be solved by relying nearly exclusively on the co-linearity of neighboring paddles whereas contour detection in natural images might oftentimes require richer, sometimes non-directional, measures of local change.

**hGRU explains human contour detection**   The patterns of horizontal connectivity such as association fields are thought to underlie human performance in contour detection [18]. Given the presence of visually similar horizontal connectivity in the hGRU, we were motivated to test its ability to explain human psychophysics data during contour detection. We did this by recreating synthetic datasets from [5] that were used to measure human contour detection performance. The task involved detecting a contour of co-linearly aligned paddles in an array of randomly oriented distractor paddles. Different versions of the task were generated by varying inter-paddle distance within a contour (5 conditions).

Figure S6: An hGRU trained on contour detection in natural images and then fine-tuned on a classic contour detection task performs similarly to human observers. While the hGRU is overall more accurate than humans, the performance of both is strained as the distance between paddles increases. Inlays show example contour stimuli with varying distance between their paddles (from left to right: larger-to-smaller gaps). Note that we use red to highlight contour stimuli for visualization, but it is absent during the task.

We generated 1,000,000 unique $150 \times 150$ pixel images for every condition (28 total). In each case, 90% of images (900,000) were used for training and the remaining 10% (100,000) for testing. We fine-tuned our BSDS-trained hGRU separately on each of these training datasets and recorded its accuracy on the corresponding test datasets for comparison with human observers.

Although the hGRU was accurate at detecting paddles when the distance between paddles increased, this manipulation still strained network performance. We compared hGRU performance with human participants, whose responses were digitally extracted from [5] and averaged together for every experimental condition. Plotting the accuracy of the hGRU against the reported "detection score" of human observers revealed similar straining for both in response to increasing inter-paddle distance (Fig. S6).