[Reviews · NeurIPS 2018]

Reviewer 1



* Update after author response* My main request for improving the comparison with state of the art models has been done and turned out favorable for the authors‘ model. Therefore I think the paper makes a good contribution despite a number of smaller issues, most of which I hope the authors will fix in the final version. I can’t quite follow R2‘s criticism, as the authors‘ GRU baseline does essentially what they describe as a 2D GRU. *Original review* Summary: The authors introduce a novel neural network architecture called the horizontal gated recurrent unit (hGRU), which they show excels at a visual task that requires detecting related features over long spatial ranges. This task – the Pathfinder Challenge – has been used in the neuroscience literature and requires deciding whether two dots in an image are connected by a path made of short line segments. They show that a single layer of hGRU can solve this task almost perfectly, while CNNs need to be quite deep to achieve comparable performance and require orders of magnitude more parameters. Strengths: + The paper is well motivated and conceptually very clear. + The Pathfinder challenge uses simple images to generate a non-trivial and interesting task. + The paper shows a limitation of CNNs and proposes an effective solution using a gated recurrent model. + The result that a one-layer recurrent model can solve the task is quite remarkable. + Ablation studies and comparisons with other models show that the proposed hGRU model maximizes the ratio of performance to the number of parameters. Weaknesses: - The hGRU architecture seems pretty ad-hoc and not very well motivated. - The comparison with state-of-the-art deep architectures may not be entirely fair. - Given the actual implementation, the link to biology and the interpretation in terms of excitatory and inhibitory connections seem a bit overstated. Conclusion: Overall, I think this is a really good paper. While some parts could be done a bit more principled and perhaps simpler, I think the paper makes a good contribution as it stands and may inspire a lot of interesting future work. My main concern is the comparison with state-of-the-art deep architectures, where I would like the authors to perform a better control (see below), the results of which may undermine their main claim to some extent. Details: - The comparison with state-of-the-art deep architectures seems a bit unfair. These architectures are designed for dealing with natural images and therefore have an order of magnitude more feature maps per layer, which are probably not necessary for the simple image statistics in the Pathfinder challenge. However, this difference alone increases the number of parameters by two orders of magnitude compared with hGRU or smaller CNNs. I suspect that using the same architectures with smaller number of feature maps per layer would bring the number of parameters much closer to the hGRU model without sacrificing performance on the Pathfinder task. In the author response, I would like to see the numbers for this control at least on the ResNet-152 or one of the image-to-image models. The hGRU architecture seems very ad-hoc. - It is not quite clear to me what is the feature that makes the difference between GRU and hGRU. Is it the two steps, the sharing of the weights W, the additional constants that are introduced everywhere and in each iteration (eta_t). I would have hoped for a more systematic exploration of these features. - Why are the gain and mix where they are? E.g. why is there no gain going from H^(1) to \tilde H^(2)? - I would have expected Eqs. (7) and (10) to be analogous, but instead one uses X and the other one H^(1). Why is that? - Why are both H^(1) and C^(2) multiplied by kappa in Eq. (10)? - Are alpha, mu, beta, kappa, omega constrained to be positive? Otherwise the minus and plus signs in Eqs. (7) and (10) are arbitrary, since some of these parameters could be negative and invert the sign. - The interpretation of excitatory and inhibitory horizontal connections is a bit odd. The same kernel (W) is applied twice (but on different hidden states). Once the result is subtracted and once it's added (but see the question above whether this interpretation even makes sense). Can the authors explain the logic behind this approach? Wouldn't it be much cleaner and make more sense to learn both an excitatory and an inhibitory kernel and enforce positive and negative weights, respectively? - The claim that the non-linear horizontal interactions are necessary does not appear to be supported by the experimental results: the nonlinear lesion performs only marginally worse than the full model. - I do not understand what insights the eigenconnectivity analysis provides. It shows a different model (trained on BSDS500 rather than Pathfinder) for which we have no clue how it performs on the task and the authors do not comment on what's the interpretation of the model trained on Pathfinder not showing these same patterns. Also, it's not clear to me where the authors see the "association field, with collinear excitation and orthogonal suppression." For that, we would have to know the preferred orientation of a feature and then look at its incoming horizontal weights. If that is what Fig. 4a shows, it needs to be explained better.

Reviewer 2



Summary The authors introduce an image classification model designed for images which have non-local features, i.e. features with long spatial distance. Their recurrent neural network, called hGRU, is said to be inspired by neuroscience theories on cortical circuits, but based on an LSTM variant called GRU [17]. They provide empirical results on the Pathfinder dataset, where two white dots on a black background are either connected by white dashed lines or not. The authors compare to a variety of other models. While several models seem to achieve near 100% test accuracy, hGRU does so with 200 times fewer trainable parameters than the nearest competitor. Finally, they compare and, to some extent, analyse human scores and a trained hGRU on a contour detection task in a natural scene. Disclaimer: This review will focus mostly focus on the Machine Learning aspect since we have little expertise in visual cortical circuits. The paper correctly identifies known limitations of CNNs and proposes a solution given the current state of knowledge regarding visual cortical circuits. This may be a strength of the paper as it links RNN machine learning models to such theories. For the Pathfinder task, the hGRU seems to show good empirical results. One weakness of this paper is its lack of clarity and aspects of the experimental evaluation. The ResNet baseline seems to be just as good, with no signs of overfitting. The complexity added to the hGRU model is not well motivated and better baselines could be chosen. What follows is a list 10 specific details that we would like to highlight, in no particular order: 1. Formatting: is this the original NIPS style? Spacing regarding sections titles, figures, and tables seem to deviate from the template. But we may be wrong. 2. The general process is still not 100% clear to us. The GRU, or RNNs in general, are applied to sequences. But unlike other RNNs applied to image classification which iterate over the pixels/spatial dimensions, the proposed model seems to iterate over a sequence of the same image. Is this correct? 2.1 Comment: The general high-level motivation seems to be map reading (see fig 1.c) but this is an inherently sequential problem to which we would apply sequential models so it seems odd that one would compare to pure CNNs in the first place. 3. Section 2 begins with a review of the GRU. But what follows doesn't seem to be the GRU of [17]. Compare eq.1 in the paper and eq.5 in [7]. a) there doesn't seem to be a trained transformation on the sequence input x_i and b) the model convolves the hidden state, which the standard GRU doesn't do (and afaik the convolution is usually done on the input stream, not on the hidden state). c) Since the authors extend the GRU we think it would make section 2 much more readable if they used the same/similar nomenclature and variable names. E.g., there are large variations of H which all mean different things. This makes it difficult to read. 4. It is not clear what horizontal connections are. One the one hand, it seems to be an essential part of the model, on the other hand, GRU is introduced as a method of learning horizontal connections. While the term certainly carries a lot of meaning in the neuroscience context, it is not clear to us what it means in the context of an RNN model. 5. What is a feed forward drive? The equations seem to indicate that is the input at every sequence step but the latter part of the sentence describes it as coming from a previous convolutional layer. 6. The dimensions of the tensors involved in the convolution don't seem to match. The convolution in a ConvNet is usually a 2D discrete convolution over the 2 spatial dimensions. If the image is WxHxC (width, height, and, e.g., the 3 colour channels), and one kernel is 1x1xC (line 77) then we believe the resulting volume should be WxHx1 and the bias is a scalar. The authors most certainly want to have several kernels and therefore several biases but we only found this hyper-parameter for the feed forward models that are described in section 3.4. The fact that they have C biases is confusing. 7. Looking very closely at the diagram, it seems that the ResNet architectures are as good if not even slightly better than the hGRU. Numerical measurements would probably help, but that is a minor issue. It's just that the authors claim that "neural networks and their extensions" struggle in those tasks. Since we may include ResNets in that definition, their own experiment would refute that claim. The fact that the hGRU is using many fewer parameters is indeed interesting but the ResNet is also a more general model and there is (surprisingly) no sign of overfitting due to a large model. So what is the motivation of the authors of having fewer parameters? 8. Given the fact that ResNets perform so well on this task, why didn't the authors consider the earlier and closely related highway (HW) networks [high1]? HWs use a gating mechanism which is inspired by the LSTM architecture, but for images. Resnets are a special case of HW, that is, HW might make an even stronger baseline as it would also allow for a mix and gain-like computation, unlike ResNets. 9. In general, the hGRU is quite a bit more complex than the GRU. How does it compare to a double layer GRU? Since the hGRU also introduces a two-layer like cell (inhibiton part is seperated by a nonlinearity from the exhibition part) it seems unfair to compare to the GRU with fewer layers (and therefore smaller model complexity) 10. Can the authors elaborate on the motivation behind using the scalars in eq 8-11? And why are they k-dimensional? What is k? 11. Related work: The authors focus on GRU, very similar to LSTM with recurrent forget gates [lstm2], but GRU cannot learn to count [gru2] or to solve context-free languages [gru2] and also does not work as well for translation [gru3]. So why not use "horizontal LSTM" instead of "horizontal GRU"? Did the authors try? What is the difference to PyramidLSTM [lstm3], the basis of PixelRNNs? Why no comparison? Authors compare against ResNets, a special case of the earlier highway nets [high1]. What about comparing to highway nets? See point 8 above. [gru2] Weiss et al. On the Practical Computational Power of Finite Precision RNNs for Language Recognition. Preprint arXiv:1805.04908. [gru3] Britz et al (2017). Massive Exploration of Neural Machine Translation Architectures. Preprint arXiv:1703.03906 [lstm2] Gers et al. “Learning to Forget: Continual Prediction with LSTM.“ Neural Computation, 12(10):2451-2471, 2000. [lstm3] Stollenga et al. Parallel Multi-Dimensional LSTM, With Application to Fast Biomedical Volumetric Image Segmentation. NIPS 2015. Preprint: arxiv:1506.07452, June 2015. [high1] Srivastava et al. Highway networks. Preprints arXiv:1505.00387 (May 2015) and arXiv:1507.06228 (Jul 2015). Also at NIPS'2015. After the rebuttal phase, this review was edited by adding the following text: Thanks for the author feedback. However, we remain unconvinced. The baseline methods used for performance comparisons (on a problem on which few compete) are not the state of the art methods for such tasks - partially because they throw away spatial information the deeper they get, while shallower layers cannot connect the dots (literally) due to the restricted field of view. Why don't the authors compare to a state of the art baseline method that can deal with arbitrary distances between pixels - standard CNNs cannot, but the good old multi-dimensional (MD) RNN can (https://arxiv.org/abs/0705.2011). For each pixel, a 2D-RNN implicitly uses the entire image as a spatial context (and a 3D-RNN uses an entire video as context). A 2D-RNN should be a natural competitor on this simple long range 2D task. The RNN is usually LSTM (such as 2D-LSTM) but could be something else. See also MD-RNN speedups through parallelization (https://arxiv.org/abs/1506.07452). The submission, however, seems to indicate that the authors don’t even fully understand multi-dimensional RNNs, writing instead about "images transformed into one-dimensional sequences” in this context, although the point of MD-RNNs is exactly the opposite. Note that an MD-RNN in general does have local spatial organization, like the model of the authors. For any given pixel, a 2D-RNN sees this pixel plus the internal 2D-RNN states corresponding to neighbouring pixels (which already may represent info about lots of other pixels farther away). That’s how the 2D-RNN can recursively infer long range information despite its local 2D spatial neighbourhood wiring. So any good old MD-RNN is in fact strongly spatially organised, and in that sense even biologically plausible to some extent, AFAIK at least as plausible as the system in the present submission. The authors basically propose an alternative local 2D spatial neighbourhood wiring, which should be experimentally compared to older wirings of that type. And to our limited knowledge of biology, it is not possible to reject one of those 2D wirings based on evidence from neuroscience - as far as we can judge, the older 2D-RNN wiring is just as compatible with neurophysiological evidence as the new proposal. Since the authors talk about GRU: they could have used a 2D-GRU as a 2D-RNN baseline, instead of their more limited feedforward baseline methods. GRU, however, is a variant of the vanilla LSTM by Gers et al 2000, but lacking one gate, that’s why it has those problems with counting and with recognising languages. Since the task might require counting, the best baseline method might be a 2D-LSTM, which was already shown to work on challenging related problems such as brain image segmentation where the long range context is important (https://arxiv.org/abs/1506.07452), while I don’t know of similar 2D-GRU successes. We also agree with the AC regarding negative weights. Despite some motivation/wording that might appeal to neuroscientists, the proposed architecture is a standard ML model that has been tweaked to work on this specific problem. So it should be compared to the most appropriate alternative ML models (in that case 2D-RNNs). For now, this is a Machine Learning paper slightly disguised as a Computational Neuroscience paper. Anyway, the paper has even more important drawbacks than the baseline dispute. Lack of clarity still makes it hard to re-implement and reproduce, and a lot of complexity is added which is not well motivated or empirically evaluated through, say, an ablation study. Nevertheless, we encourage the authors to produce a major revision of this interesting work and re-submit again to the next conference!

Reviewer 3



This paper proposes a novel neural network module called a horizontal gated-recurrent unit. The authors motivate it via a nicely explained and referenced appeal to the neuroscience of visual cortex and the psychophysics of contour detection. The proposed model is an extension of a convolutional GRU model to add long-range horizontal excitatory and inhibitory connections. One notable property is that it has fewer free parameters than other comparable models, this is nicely shown in Fig. 3C, a type of plot comparing approaches by their number of free parameters that I would be happy to see more frequently in NIPS papers. I found this paper nicely constructed and convincing. I think this model could be useful in computational neuroscience as well as machine learning. I'm happy to recommend its publication at NIPS. I have one question, it was mentioned in the human data section at the top of page 8 that humans, unlike the model, were sensitive to the length of the path. Do you think this indicates the model is doing something different from the way the humans are solving it? What about human reaction time, I suppose that it scales with the length of the path. This model has a notion of time, so it seems like such effects could be addressed with it. Do you think it would work, i.e. that you could model reaction time effects this way? Minor: what is meant by "digitally extracted" in line 283 (near top of page 8)?